# Development of Nested PCR for SARS-CoV-2 Detection and Its Application for Diagnosis of Active Infection in Cats

**DOI:** 10.3390/vetsci9060272

**Published:** 2022-06-05

**Authors:** Ivo Sirakov, Ralitsa Popova-Ilinkina, Dobrinka Ivanova, Nikolina Rusenova, Hristiyan Mladenov, Kalina Mihova, Ivan Mitov

**Affiliations:** 1Department of Medical Microbiology, Medical Faculty, Medical University, 1431 Sofia, Bulgaria; docbiba@hotmail.com (D.I.); igmitov@gmail.com (I.M.); 2GenLab Ltd., 8000 Burgas, Bulgaria; r.ilinkina@gmail.com; 3Second City Hospital Sofia, 1202 Sofia, Bulgaria; 4Department of Veterinary Microbiology, Infectious and Parasitic Diseases, Faculty of Veterinary Medicine, Trakia University, 6000 Stara Zagora, Bulgaria; n_v_n_v@abv.bg; 5Diagnostic Consulting Center 14, 1408 Sofia, Bulgaria; hristian_m@yahoo.com; 6Molecular Medicine Center, Department of Medical Chemistry and Biochemistry, Medical Faculty, Medical University, 1431 Sofia, Bulgaria; kalina_mihova@abv.bg

**Keywords:** nested PCR, SARS-CoV-2, humans, cats, validation

## Abstract

SARS-CoV-2 emerged in 2019 and found diagnostic laboratories unprepared worldwide. To meet the need for timely and accurate virus detection, laboratories used rapid Ag tests and PCR kits based on costly multi-channel real-time techniques. This study aimed to develop a conventional nested PCR based on the SARS-CoV-2 N gene, validate it against some approved assays, and apply it to samples from six cats with respiratory symptoms obtained in early 2020 during the first COVID-19 wave in humans in Bulgaria. The nested PCR technique showed 100% sensitivity and specificity; it could detect extracted SARS-CoV-2 RNA at concentrations as low as 0.015 ng/μL. The results identified the six tested cat samples as positive. Sequence analysis performed in two of them confirmed this. The presented technique is reliable, easy to implement and inexpensive, and can be successful in strategies for the prevention and control of SARS-CoV-2 in humans, cats and other susceptible species.

## 1. Introduction

Evidence has shown that coronaviruses infecting humans are of animal origin [1]; thus, understanding their zoonotic nature is a key factor in combating them [2]. Humans—with their social, cultural and economic activities—play a major role in cross-species virus transmission [3]. Although researchers have associated SARS-CoV-2 with bats [4], some suggest it has originated from unknown zoonotic events [5]; the mechanism and all species implicated in its circulation in nature remain unclear. The SARS-CoV-2 pandemic causes serious healthcare, social, economic and personal issues. To find solutions to these challenges, it is important to understand the virus’ nature, through all its characteristics at present.

The host cell receptors that interact with SARS-CoV-2—ACE2 [6] and GRP78 [7]—show differential expression in different organs in humans and animals. Researchers found various animals susceptible to this virus: pigs, dogs, ducks, chickens and green monkeys; with ferrets and cats being permissive to infection [8,9]. Some of these or other species may become a reservoir of infection [10]. Thus, it is important to focus research on this aspect, as well as on differential diagnoses of other infectious and/or respiratory diseases in these animals.

Routine SARS-CoV-2 diagnosis in humans employs mainly in vitro diagnostics (IVD) real-time reverse transcription (RT) PCR kits based on several virus genes, whereas other researchers have developed nested PCR based on the nonstructural ORF1ab gene [11]. There are no other standardized techniques for routine SARS-CoV-2 diagnosis in other susceptible species. Nested PCR has very high sensitivity, and laboratories use both its conventional and real-time versions for the early diagnosis of oncological diseases, too [12].

This prompted us to design a conventional nested PCR for SARS-CoV-2 diagnosis during the first wave of infection in Bulgaria, which began in March 2020. We applied the technique to test six cats with respiratory symptoms. The aim of the study was to validate the developed nested PCR technique against some approved IVD SARS-CoV-2 assays and to perform sequence analyses for accurate diagnosis of suspected cases of SARS-CoV-2 infection in humans and cats in Bulgaria.

## 2. Materials and Methods

### 2.1. Samples

To validate the technique, we used 45 SARS-CoV-2 positive samples from symptomatic patients and 45 negative samples from clinically healthy individuals collected at a local clinic (Diagnostic and Consulting Center 14, Sofia, Bulgaria) and Second General Hospital for Active Treatment (Sofia, Bulgaria). The samples were collected from July 2020 to September 2021. The study also included six symptomatic cats: swabs (conjunctival, nasal and oral) were collected at GenLab Laboratory (Burgas, Bulgaria) during the first wave, in March–April 2020. The samples were analyzed twice: at the time of collection and 20 months later. Samples of extracted RNA and cDNA were stored at −80 °C. The second analysis (at 20 months) was performed on the initial samples.

To test the specificity of the primers (Table 1), we used cDNA and DNA from samples positive for other cat pathogens: *Feline herpesvirus* (FHV) positive DNA (lab code GL6v/2019), *Feline calicivirus* (FCV) positive cDNA (lab code 109/Pr2021, also positive for *Chlamydia* and *Mycoplasma* spp.), *Feline coronavirus* (FCoV) positive cDNA (lab code GL195/2016, also positive for *Chlamydia*)*,* and the *Chlamydia-Mycoplasma* spp. positive sample (lab code 107/Pr2021). Detection of FHV, *Chlamydia* and FCV was done according to Sykes et al. [13], FCoV [14] and *Mycoplasma* spp. [15].

To determine the sensitivity of the nested PCR, we used control of 10 μL, 4.5 × 10^9^ genome copies per mL, inactivated isolate SARS-CoV-2 USA/WA1/2020 (Microbiologics, Saint Cloud, MN, USA).

### 2.2. Reference Method for Detection of SARS-CoV-2

The reference methods in this study were a loop-mediated isothermal amplification (LAMP) IVD test Ender Mass (Switzerland), a LiliF COVID-19 real-time RT-PCR kit (LiliF Diagnostics, iNtRON Biotechnology, Seongnam, Korea) and a real-time multiplex RT-PCR kit (Labsystems Diagnostics Oy, Vantaa, Finland), according to the manufacturers’ instructions.

### 2.3. RNA Extraction and Nested PCR

Amplification reactions were run following RNA extraction using the ISOLATE II RNA Mini kit (Bioline, Meridian Bioscience, Memphis, TN, USA). For LAMP IVD, we used the buffer included in the kit according to the instructions.

We performed reverse transcription (RT) using the SensiFAST cDNA synthesis kit (Bioline, Meridian Bioscience, Memphis, TN, USA): 7 μL of extracted RNA, 8 μL of DEPC treated water, 4 μL of TransAmp buffer and 1 μL of RT enzyme. The reaction conditions were: 25 °C—10 min, 42 °C—15 min, 80 °C—5 min. Storage was at −10 °C. The nested PCR primers are shown in Table 1. The first round of nested PCR used a reaction volume of 25 μL: 12.5 μL of My Taq HS red mix (Bioline, Meridian Bioscience, Memphis, TN, USA), 4 μL of cDNA, 1 μL of each external primer (10 pmol/μL each) and 6.5 μL of PCR grade water (Bioline, Meridian Bioscience, Memphis, TN, USA). The reaction volume in the second round was 25 μL: 12.5 μL of My Taq HS red mix (Bioline, Meridian Bioscience, Memphis, TN, USA), 0.5 μL of the first PCR, 1 μL of each internal primer (10 pmol/μL each) and 10 μL of PCR grade water (Bioline, Meridian Bioscience, Memphis, TN, USA).

To test the sensitivity, we ran amplification reactions with different quantities of RNA extracted from the SARS-CoV-2 reference strain. To test the primer specificity, the reaction volume was 25 μL: 12.5 μL of My Taq HS red mix (Bioline, Meridian Bioscience, Memphis, TN, USA), 5 μL of cDNA or DNA (positive for microorganisms causing similar clinical symptoms in cats, as described above), 1 μL of each primer (10 pmol/μL each) and 5.5 μL of PCR grade water (Bioline, Meridian Bioscience, Memphis, TN, USA). The thermocyclers for the amplification reactions were QB96 (Quanta Biotech, Surrey, UK), SaCycler-96 (Sacace Biotechnologies, Como, Italy) and FluoroCycler (Hain Lifescience, Nehren, Germany).

### 2.4. Sequencing

We sequenced two of the positive samples from cats using the internal primers, as follows. Aliquots of 1.5 μL from each amplified sample were purified enzymatically to degrade the residual primers and nucleotides using the Exo-CIP Rapid PCR Cleanup Kit (New England Biolabs, Ipswich, MA, USA), according to the manufacturer’s instructions. The sequencing reaction was performed with a forward and reverse primer using the Big Dye Terminator kit, v3.1 (Applied Biosystems, Bedford, MA, USA), according to the manufacturer’s instructions. Residual labeled nucleotides and primers were removed by EDTA/sodium citrate/ethanol precipitation. Sequencing results were read on an automated capillary sequencing instrument (ABI 3500xl, Applied Biosystems, Bedford, MA, USA).

### 2.5. Qualitative and Quantitative Control of Extracted RNA and PCR Products

RNA extracts were analyzed by NanoDrop 2000 (Thermo Fisher Scientific, Waltham, MA, USA). Gel electrophoresis was used for the quantitation of the reference SARS-CoV-2 strain and for qualitative analysis of the other PCR products. Gel electrophoresis was performed with 2% agarose (Lonza Group AG, Basel, Switzerland), 10 ng/mL of ethidium bromide (Sigma-Aldrich, Merck KGaA, Saint Louis, MO, USA), 1×TAE buffer, and 1 kb DNA ladder (Bioline, Meridian Bioscience, Memphis, TN, USA) at 120–150 V, 70–120 mA for 30 min. The gel was visualized using a UV transilluminator (Biobase, Jinan, China) at 240/260 nm.

We calculated the specificity and sensitivity of the technique against the IVD assays as described [16]. The software for primer and sequence processing and analysis was MegaX [17] and blast NCBI (National Center for Biotechnology Information, Bethesda, MD, USA).

## 3. Results

Nested PCR analysis of 45 samples that tested SARS-CoV-2 negative in the IVD assays did not give any positive results.

The PCR amplification of 45 samples that were SARS-CoV-2 positive in the reference assays (IVD real-time and LAMP) produced 31 positives and 14 negatives with the external primers. The amplification of these 14 samples with the internal primers in the second round produced 212-bp products specific for SARS-CoV-2 (Figure 1a), identifying them as SARS-CoV-2 positive, too. There was 100% agreement between the results (specificity and sensitivity) calculated according to Samad et al. [16] in the IVD assays and the nested PCR described here. When we tested the specificity of the primers against samples from cats that were positive for FHV, FCV, FCoV, *Chlamydia* and *Mycoplasma* spp., there were no non-specific reactions in the first round of PCR. The second round gave a weak non-specific band of about 400 bp in two of the samples (Figure 1a, lanes 9–10, top). The purified virus RNA extracted from the reference strain was 5.7 ng/μL. We used the following quantity of virus RNA in the amplification reactions: 45.6–0.3 ng or, respectively, from 2.28 ng/μL to a 0.015 ng/μL final concentration in the reaction mixture. The first round of amplification gave positive results of up to 0.8 ng, and the second one, up to 0.3 ng (Figure 1b).

The six samples from the cats with respiratory symptoms were positive for SARS-CoV-2 in the first analysis: four samples with the external primers and two samples with the internal primers (data not shown). The second analysis, 20 months later, after validation of the technique, confirmed five of the samples, all with the internal primers. One of the samples gave a negative result (Figure 1a).

Next, we applied the SARS-CoV-2 IVD assays used for human patients to test the samples from the cats that were positive in the nested PCR. In the real-time RT-PCR, a curve appeared after 33 cycles and had a small slope (Figure 2), however, LAMP did not confirm the samples.

The analysis of the processed 212-bp sequences using blast NCBI showed 100% identity with 100 SARS-CoV-2 sequences in the GenBank database, NCBI. The sequences are deposited in GenBank, NCBI, ref. no. OM038466 and OM038614.

## 4. Discussion

SARS-CoV-2 diagnosis has been mainly based on real-time RT-PCR kits, which involve specialized training and equipment requirements. Conventional, nested PCR is easy, inexpensive, does not require interpretation and has very high sensitivity [12]. The nucleoprotein (N) gene is one of the most conservative SARS-CoV-2 genes. Its conservative nature stems from the key functions that the protein it encodes has [18], and hence, the lower possibility for mutations to accumulate as the virus adapts to changes in its environment: immune response or therapy. Sequence analysis of SARS-CoV-2 isolates from various species of animals: cats, dogs, minks, mice and a tiger, identified a mutation in the region that we selected for amplification [19]. The mutation is in the codon encoding amino acid 80 (238–240 nt) in minks (all mutations in the N gene are detectable in this species) [19]. Since this mutation is external to the nucleotide sequences targeted by our primers, it will not affect the sensitivity of the technique. However, the mutation may become important in epidemiological studies in case it proves to be a marker for species-specific virus strains in minks. This mutation was not present in our sequences.

Since RNA extraction kits have various specifics, and 100% elution is difficult to achieve, we diluted the extracted RNA rather than the initial virus sample. The results from testing SARS-CoV-2 positive and negative samples, and a range of dilutions of reference virus RNA, showed that the technique is highly sensitive and specific. In the specificity assay, the non-specific bands in the nested PCR, from complementary DNA, in two of the samples were larger than expected and did not interfere with the interpretation of the results.

The inconsistent results for the samples from the six cats in the two tests, before and after the validation of the technique, most likely resulted from the long storage at two different temperatures and/or the transportation, leading to nucleic acid degradation [20]. Despite some disadvantages of the RT-PCR as false-positive and false-negative results, numerous steps and the need for specific lab equipment, it has become a gold standard molecular diagnostic technique for COVID-19 [21,22,23]. In addition, the RT-PCR results may be influenced by the type of collected specimen, the clinical symptoms of the patient and the period of time between sampling and the onset of symptoms [24,25]. Regarding nested PCR, whether conventional or real-time, cross-contamination of the samples was outlined as a drawback [26]. However, we deem that when the aseptic working principles in a virology laboratory are observed, the risk of contamination is minimized. This was confirmed both in this and in previous research of ours [27].

Sequencing of the PCR amplification products from the samples from cats that tested SARS-CoV-2 positive in the nested PCR, confirmed the specificity of the technique, and consequently, the active SARS-CoV-2 infection in these animals during the first wave in Bulgaria.

The real-time RT-PCR results for the five samples from cats that tested positive are open to discussion and interpretation. One possibility could be a very low concentration of nucleic acid (RNA and cDNA), nearly reaching the method’s limit of detection. The negative results for the positive samples from cats in the LAMP assay could have resulted from mutations in the target region (ORF1a) that occur in SARS-CoV-2 isolates from various animal species [19]. Another possible reason is the extraction procedure. The extraction step has been validated for human samples, however, those from cats may contain inhibitory substances. These limitations are not present in our procedure. The authors report multiple (32 non-synonymous) mutations in ORF1ab in all isolates from cats, dogs, minks, mice and a tiger. Such variability suggests the speculation that other future mutations may occur in this gene along with virus transmission to various hosts, thus decreasing its diagnostic sensitivity.

Cats are reportedly susceptible and can shed SARS-CoV-2 [8,28]. Our results also confirmed that cats are susceptible. Therefore, we recommend efforts toward combating the pandemic to include routine SARS-CoV-2 diagnostics in cats. This would be of benefit for differential diagnosis of other pathogens causing similar symptoms in cats: FHV, FCV, *Chlamydia* and *Mycoplasma* spp.

## 5. Conclusions

The conventional nested PCR designed here for the detection of SARS-CoV-2 showed 100% sensitivity and specificity against the IVD techniques used, real-time RT-PCR and LAMP. It needs inexpensive equipment and allows diagnosis within 2.5–6 h. It can expand the range of studies on the spread of SARS-CoV-2 among susceptible species other than humans. This could contribute to a better understanding of the circulation of SARS-CoV-2 and, in turn, could widen the strategies for its prevention and control.

## Figures and Tables

**Figure 1 vetsci-09-00272-f001:**
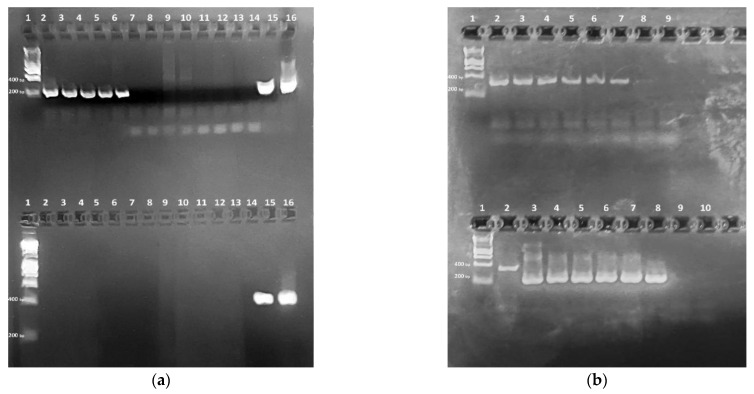
Nested PCR amplification products separated via electrophoresis in agarose gel. (**a**) Specificity of nested PCR assay determined using six cDNA samples from cats. 1—DNA Ladder 1 kb (Bioline, Meridian Bioscience, Memphis, TN, USA); 2–7—samples from cats with respiratory symptoms; 8—negative control. Determination of the specificity of nested PCR assay. 9—FCoV positive sample; 10—FCV positive; 11—*Chlamydia* and *Mycoplasma* spp. positive; 12—FHV positive; 13—IVD real-time RT-PCR SARS-CoV-2 negative human sample; 14—IVD LAMP SARS-CoV-2 negative human sample; 15—IVD real-time RT-PCR SARS-CoV-2 positive human sample; 16—IVD LAMP SARS-CoV-2 positive human sample. Bottom line I: external primers, expected size of product 335 bp and; top line II: internal primers, expected size of products 212 bp. (**b**) Sensitivity of nested PCR assay determined using different quantities of extracted RNA from inactivated SARS-CoV-2 USA/WA1/2020 isolate. 1—DNA Ladder 1 kb; top line, external primers: 2—45.6 ng (final concentration in the reaction mixture 2.28 ng/μL), 3—11.4 ng (0.57 ng/μL), 4—5.7 ng (0.285 ng/μL), 5—3.7 ng (0.185 ng/μL), 6—1.3 ng (0.065 ng/μL), 7—0.8 ng (0.04 ng/μL), 8—0.3 ng (0.015 ng/μL), 9—negative control of external primers. Bottom line internal primers, 2—external primers, 9—first negative control (from the first reaction) primers; 10—negative control of internal primers.

**Figure 2 vetsci-09-00272-f002:**
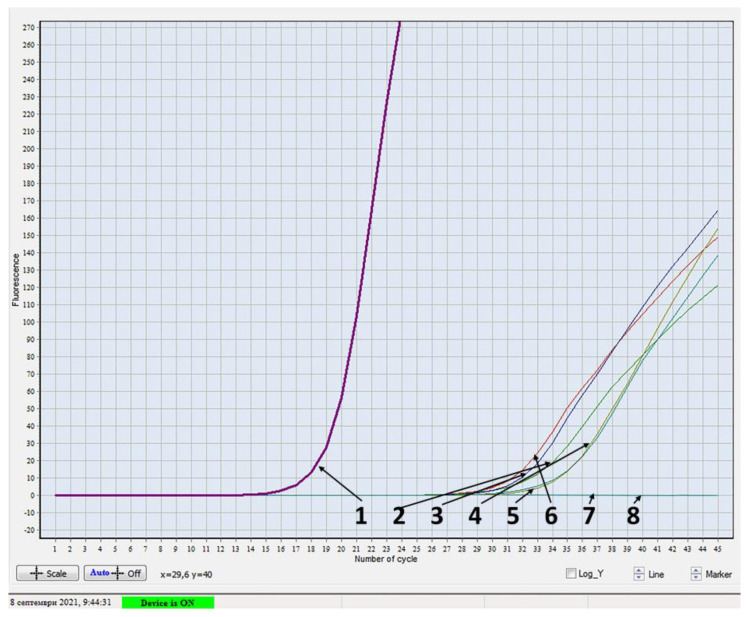
Results of six samples from cats by using real-time RT-PCR 1−positive control; 2−7−samples from cats with respiratory symptoms; 8−negative control.

**Table 1 vetsci-09-00272-t001:** Nested PCR primers for detection of the nucleocapsid phosphoprotein (N) gene of SARS-CoV-2.

Primer	Sequence (5′–3′)	Position ^a^	Annealing ^b^ (T°C)	Size (bp)
Ext2019nCorVF	GGCAGTAACCAGAATGGAGA	28346–28365	54.6	335
Ext2019nCorVR	CTCAGTTGCAACCCATATGAT	28681–28661		
intF	CACCGCTCTCACTCAACAT	28432–28450	54.6	212
intR	CATAGGGAAGTCCAGCTTCT	28643–28624		

^a^ The primers were designed according to the position of the N gene in the SARS-CoV-2 genome, sequence MN908947.3, isolate Wuhan-Hu-1: 28274–29533 [4]. ^b^ Annealing temperature according to QB96 software (Quanta Biotech, Surrey, UK).

## Data Availability

The data presented in this study can be found in the manuscript.

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
