# Peer review of "Development of Nested PCR for SARS-CoV-2 Detection and Its Application for Diagnosis of Active Infection in Cats"

_vetsci, 2022, doi:10.3390/vetsci9060272_

Round 1

Reviewer 1 Report

The paper of Sirakov et al. Describes nested PCR method for detection of SARS-CoV2 in human and cat samples. As in the course of COVID-19 pandemy plenty of NAT methods have been developed for SARS-CoV2 diagnosis and many of them are validated for IVD, this paper is only of moderate importance. Its main contribution is validation of the method described for SARS-CoV2 diagnostics in domestic cats. However, some sentences in both the Materials and Methods a Results paragraphs are not clearly formulated and need amendment or modification. Moreover, it is not documented, whether the method is able to detect all the SARS-CoV2 variants.

Main comments:

Materials and methods:

Line 68-69: It is not clear what kind of material was analysed repeatedly: The samples, RNA or cDNA extracts prepared from them? If the first is correct, how the samples were stored before the second analysis? Were there aliquoted or repeatedly thawed?

Results:

The sentence 128-130 should be re-formulated:  Positivity of 31/45 samples does not give 100%. Which „other“ samples? The samples negative in the first run of nested PCR?

Lines 136-138: How can be the concentration  45.5 ng/µL reached from 5.7 ng/µL of the reference strain RNA?

Discussion:

Line 171: Dilution 107 does not tell anything as for the sensitivity of the assay.

The sensitivity of the nested PCR for currently prevailing SARS-CoV2 variants should be documented.

Minor comments:

Materials and methods:

Explain abbreviations FHV,FCV. The methods used for characterization of FNV, FCV or FCoV positivity should be cited.

Line 74: What does 109  in the reference strain mean? Genome copies or the virus particles?

Results:

Line 114-115: The sentence should be divided into two: RNA extracts were probably analysed by NanoDrop, while PCR products by  gel electrophoresis ( which was used for quantitation of the reference SARS-CoV2 strain, while in other PCR products it was qualitative only).

Discussion:

Line 188: ...complementary DNA....

Which countermeasures were used to prevent false positive results due to cross-contamination?

Reviewer 2 Report

Summary:

Authors designed a nested PCR based detection assay for SARS-CoV2 and tested it on human and cat samples. Test appears efficient and specific. Overall, the manuscript is straightforward and written accordingly. However, there is a clear lack of rigor in preparing the figures. Please improve visibility and presentation to reach publication standards.

Major comments:

  • Lines 62-65: The authors used 90 patient samples total. It is important to mention approval of institute (bio)ethics committee if needed. What is the nature of the samples?

  • Lines 73-74: That sentence is poorly written making it hard to understand what the authors used as control exactly.

  • Acronyms need to be defined at their first used. Many such as (IVD, LAMP, FHV, FCOV).

  • Figure 1.B: Figure legend need improving. Right now, it is very confusing. Please make sure to add visual cues to increase visibility such as:
    • Do not reset lane number from lower part,
    • Mark ext/Int primer PCR products,
    • Visual gradient/increase of RNA concentration.

  • What is the purpose of figure 2? Right now, with the current version of the manuscript it Is very unclear what the authors are trying to argue.

Round 2

Reviewer 2 Report

The authors have updated the manuscript and addressed my comments.